# Investigating the Temporal Effects of Thermal Stress on Corticosterone Release and Growth in Toad Tadpoles

**DOI:** 10.3390/biology14030255

**Published:** 2025-03-03

**Authors:** Saeid Panahi Hassan Barough, Dillon J. Monroe, Thomas C. Clark, Caitlin R. Gabor

**Affiliations:** 1Department of Biology, Texas State University, 601 University Drive, San Marcos, TX 78666, USA; saeedpanahi@hotmail.com (S.P.H.B.); dillonmonroe@suu.edu (D.J.M.); thomas.c.clark109@gmail.com (T.C.C.); 2Department of Biology, Southern Utah University, 351 W University Blvd., Cedar City, UT 84720, USA

**Keywords:** amphibian, corticosterone, development, *Incilius nebulifer*

## Abstract

Using a water-borne hormone collection method, we investigated the release of corticosterone (CORT) metabolites in response to chronic heat exposure in Gulf Coast toad tadpoles. The tadpoles were exposed to either 32 °C or control (23 °C) temperatures for 7 days. Our results showed significant differences in CORT release rates post-treatment exposure, but the differences were minimal, indicating that chronic heat exposure was not highly stressful in this species. Instead, tadpoles at 32 °C exhibited faster growth, indicating potential benefits of higher temperatures. However, there was a trade-off, as survival was lower in the heat-exposed tadpoles.

## 1. Introduction

Amphibians are declining globally due to numerous factors, including habitat loss and fragmentation, climate change, disease, contaminants, invasive species, and their synergistic effects [1,2,3]. Among these threats, rising temperatures due to climate change and urbanization pose a significant challenge to aquatic ecosystems. Land use changes and the urban heat island effect have increased water temperatures by up to 5 °C in some areas of the northeastern United States, with documented negative effects on fish and amphibians [4,5,6]. Warmer water accelerates metabolic rates in aquatic organisms, leading to cascading effects on immune function, locomotor performance, and behavior [5,7,8,9]. Increased water temperature from climate change and urbanization can have direct effects on the physiological demands of aquatic vertebrates and indirect effects on the dynamics of the whole ecosystem where they are found [10].

The Hypothalamic–Pituitary–Adrenal axis (inter-renal in amphibians; HPI) is an evolutionarily conserved physiological pathway that responds to environmental stressors, like unpredictable changes in temperature [11]. Corticosterone (the main glucocorticoid in amphibians; CORT) is released by the HPI axis and plays a vital role in energy metabolism and homeostasis [12,13]. Short-term elevations of GCs can be advantageous by mobilizing energy stores, but unpredictable or prolonged perturbations can lead to chronic stress, which is associated with elevated or downregulated GC levels, reduced growth, suppressed immune function, and even mortality [11,14]. Chronic stress can result in either habituation, desensitization, or exhaustion of the HPI axis, leading to varied baseline and stress-induced CORT levels [14,15].

Chronic stress, often arising from sustained environmental disturbances, can have profound impacts on glucocorticoid profiles and associated fitness outcomes. Integrated measures of corticosterone profiles using non-invasive methods, such as fecal GC metabolites and water-borne and urinary GCs, provide earlier indicators of chronic stress than point measures, such as those from blood plasma [16]. Non-invasive GC monitoring methods, including the use of urine, water, skin swabs, and saliva, allow researchers to collect integrated measures of episodic changes in cumulative stress over time in small organisms without euthanizing individuals [17,18]. Such methods have proven valuable for monitoring endocrine functions in amphibians and other taxa [19,20,21,22]. For example, in Rio Grande leopard frogs (*Rana berlandieri*), water-borne CORT release rates increased 1 h post-exposure to an acute stressor (agitation stress) and began to decline after 2 h, with full recovery taking up to 6 h [13]. The integration of repeated sampling over time provides insight into the role of stress physiology in breeding, survival, and population dynamics across seasons and years [23,24]. However, the duration of the lag time between HPI axis activation and the hormonal response to chronic stressors remains uncertain when using non-invasive sampling methods [25].

Temperature is a particularly influential environmental stressor for amphibians, affecting both stress physiology and development. Higher temperatures and more frequent exposure to elevated temperatures are key outcomes of global climate change that will impact organismal physiology. Glucocorticoids are well-studied hormones that are part of the vertebrate physiological stress response and are expected to play an important role in determining vertebrate responses to the elevated temperatures associated with global climate change [26,27]. Comparative studies have demonstrated connections between temperature and glucocorticoid physiology across taxonomic groups [24,28,29,30]. Elevated temperatures are often linked to increased glucocorticoid secretion and phenotypic changes that may influence fitness [15,31]. In anurans and other ectotherms, the absence of efficient physiological thermoregulation [32] results in a strong reliance on temperature to regulate the growth and differentiation processes. Therefore, temperature can be considered, besides energy uptake, to be the most important proximal cause of variation in size and age at metamorphosis. As a result, larval anurans grown at cold temperatures have prolonged developmental periods, but they are also larger as metamorphs than conspecifics grown at warmer temperatures. This phenomenon makes up one of the most general rules for ectotherms [33]. When we consider CORT, Goff [34] reported that *Rana berlandieri* tadpoles housed at 27 °C exhibited lower CORT release rates and a higher body condition than those at 19 °C. The authors hypothesized that lower CORT levels prolonged the time to metamorphosis, allowing tadpoles to grow larger before metamorphosis. These studies underscore the complex, species-specific relationships between temperature, stress physiology, and fitness outcomes.

In this study, we examined the effects of chronic heat stress on corticosterone release rates and growth in Gulf Coast toad tadpoles (*Incilius nebulifer*), a species adapted to variable thermal environments. *Incilius nebulifer* is native to central and eastern Texas and northeast Mexico, but has been expanding its range northward and eastward in the United States [35]. This range expansion has been strongly associated with urbanized and disturbed habitats, suggesting that *I. nebulifer* may be particularly urban-tolerant [28]. Given that urban areas experience elevated temperatures due to the heat island effect, this species provides a valuable model for understanding how amphibians physiologically respond to chronic heat exposure in increasingly warming and urbanized environments. We exposed tadpoles to water temperatures maintained at 32 °C or 23 °C (control) for 7 days, then gradually returned the 32 °C treatment to 23 °C, simulating a chronic heat stress event. We collected water-borne samples 1 h, 2 h, 6 h, 24 h (Day 8), 48 h (Day 9) and 5 days (Day 12) post-treatment to explore the persistence of elevated corticosterone as individuals worked to re-establish homeostasis. Unlike blood plasma measures, which reflect immediate hormonal responses, water-borne hormone methods provide an integrated measure of corticosterone release rates as CORT is metabolized. We hypothesized that chronic exposure to heat would alter both the magnitude of the corticosterone release rate and extend the recovery time. Additionally, we hypothesized that the growth rate would increase with elevated temperatures. Our findings contribute to understanding the temporal dynamics of stress physiology using non-invasive methods and provide insights into the interplay between stress, growth, and thermal environments in a tolerant amphibian species facing climate-related challenges.

## 2. Materials and Methods

We collected 36 *I. nebulifer* tadpoles from Austin, TX (long: 30.16199° N, lat: 97.81194° W) in July 2022 and transported them to Texas State University, San Marcos, where they were acclimated for 14 days. During the acclimation period, the tadpoles were divided among two 6 L containers, kept on a 12 h day/night cycle at 23 °C (room temperature), and fed spirulina suspended in agar ad libitum. We used spring water and performed half water changes every 5 days. The tadpoles weighed between 34 and 93 mg at this stage and were in Gosner stages 30–35 [36]. After the acclimation period, the tadpoles were placed into a single 6 L container to randomize the individuals from the two containers, before being placed in individual 470 mL deli cups filled with spring water. We assigned tadpoles to one of two treatments for 7 days: higher water temperature (32 °C) and control (23 °C room temperature). Monroe et al. [28], using Worldclim data, found that the annual temperature for some of the southern populations of *I. nebulifer* is 23.2 °C, so we used that as our control. We have found surviving populations of tadpoles of *I. nebulifer* in ponds with water temperatures as high as 34 °C in Central Texas, much higher than average annual temperatures, but apparently less than the thermal maximum for this species. We set the upper limit below this to be sure we did not exceed the thermal maximum. The deli cups were then placed in a water tub with a circulating pump to keep temperatures consistent. We placed aquarium heaters at each end and in the middle of the high water temperature treatments to heat the water. Heaters were also placed in the low water temperature treatment water tubs as a control, but were not plugged in. We rotated the placement of the water tubs every other day to prevent any location effect. We continued the 12 h day/night cycle and ad libitum feeding throughout the experiment. The tadpoles were maintained in the same deli cup until Day 12 if they survived. On Day 12, 9/18 tadpoles maintained at 32 °C had survived and 16/18 survived when maintained at 23 °C.

Immediately prior to treatment exposure, we obtained a baseline water-borne CORT measure at 1000 h on Day 1. The water-borne CORT collection method has been validated for this species and many other amphibian species [25]. Following the 7 days of treatment exposure, we allowed the water in the heat treatment to return to room temperature before starting hormone collection (approx. 3 h). Then, we repeatedly measured CORT for each individual at 1 h, 2 h, and 6 h post-return to room temperature (Day 7), starting at 1000 h, and then after 24 h (Day 8), 48 h (Day 9), and 5 days (Day 12) post-return to room temperature at 1000 h. Our short-term sampling time points were based on methods from a similar study examining acute stress responses [13], and we extended the sampling period to assess potential longer term changes in corticosterone release rates. After each hormone sampling day (3 days), we measured their body mass. The hormones were collected using the water-borne method previously described [19]. Briefly, we placed each tadpole in a beaker containing 100 mL of spring water and allowed them to bathe for one hour. The water samples were stored at −20 °C for subsequent analysis. We extracted hormones from the water samples via vacuum filtration with Solid Phase Extraction (SPE) columns (SepPak Vac 3 cc/500 mg; Waters, Inc., Milford, MA, USA) followed by the evaporation of methanol with nitrogen gas [13]. Corticosterone (and metabolites) were measured using Enzyme Immuno-Assay (ELISA) kits after extraction and resuspension (No. 501320, Cayman Chemical Company, Inc., Ann Arbor, MI, USA). We read the absorbance of each sample using a spectrophotometer plate reader (Biotek ELX 800, Winooski, VT, USA) set to 405 nm. We then multiplied the final CORT concentrations (pg/mL) by the reconstitution volume (0.500 mL) and then divided by the tadpole mass (g) for a final unit of pg/g/h. The intra-plate variation ranged from 0.65 to 4.95% and the inter-plate variation was 8.45% for six plates. We removed the values for samples with a CV over 30%. The sample sizes ranged from 7 to 16 individuals per time point and treatment, owing to removing some samples with a high CV and mortality that occurred after Day 7 (11 died by Day 12).

To examine the effect of elevated water temperatures on CORT release rates and the mass of tadpoles, we used separate generalized linear mixed effect models (GLMM). We examined natural log-transformed CORT release rates standardized by mass (pg/g/h), with treatment (23 °C or 32 °C) and time as the main effects. We also examined body mass with treatment and time as the main effects. Following the CORT release rate analyses, we used Levene’s test to explore whether the variances were equal across treatments to explore the role of heterogeneity in variance. We also used Welch’s test to test whether the CORT release rates differed across treatments for each time point. We used a Pearson Chi square test to examine the effect of heat on survival by Day 12. We used JMP Pro 17 (SAS Institute, Inc., Cary, NC, USA).

## 3. Results

There was no significant difference in CORT release rates before we exposed the tadpoles to higher or lower temperatures on Day 1 (t = 1.418, df = 24.055, *p* = 0.169; Figure 1). We found a significant interaction between time and treatment (F_6, 147_ = 2.96, *p* = 0.009; Figure 1). We found that CORT release rates were the lowest in both treatment groups on Day 7—6 h after the first measure of CORT post-return to room temperature. Between the treatment groups, on Day 7, the 6 h CORT release rates were significantly lower in the heat-exposed treatment group, but on Day 8 the relationship switched. There was no significant difference in the variances between treatments except on Day 8 (Levene’s test *p* > 0.05). On Day 8, both treatments returned to pre-treatment levels, but the variances significantly differed (Levene’s test: F = 9.50, *p* = 0.0045). Welch’s *t*-test indicated that the CORT release rates differed significantly between the treatments on Day 7 at 6 h (F = 4.44, df = 18.16, *p* = 0.0494) and Day 8 (F = 6.67, df = 23.29, *p* = 0.0166), but no other time points showed significant differences.

We found a significant time-by-treatment interaction, where tadpoles started at the same mass, but those exposed to 32 °C gained mass faster than the tadpoles exposed to 23 °C for 7 days (F_2,59_ = 18.35, *p* < 0.001; Figure 2). The tadpoles that were exposed to 32 °C continued to grow at a faster rate for 5 days after they were returned to 23 °C. However, significantly more tadpoles in the heat treatment group were dead by Day 12 (Pearson test, χ^2^ = 6.42, *p* = 0.013).

## 4. Discussion

Given the ongoing global decline of amphibians, understanding how chronic exposure to elevated temperatures influences stress physiology and development is essential, particularly in species adapted to variable thermal environments like *Incilius nebulifer*. Using a non-invasive water-borne hormone method, we examined CORT release rates over time, following seven days of exposure to either 32 °C (heat treatment) or 23 °C (control) temperatures. Water-borne methods provide an integrated measure of CORT release rates, allowing the assessment of the persistence of elevated CORT and the recovery time as individuals re-establish homeostasis. We found that tadpoles from the control and heat treatments showed a decline in CORT release rates by approximately 6 h after our initial CORT measure, but the heat treatment group had lower CORT release rates at 6 h and higher release rates on Day 8. However, both groups ultimately returned to baseline levels within a similar timeframe. Interestingly, tadpoles exposed to elevated temperatures demonstrated significantly greater growth, both during the heat treatment and after being returned to cooler conditions. These results suggest that chronic heat exposure may promote metabolic or physiological processes that enhance growth. Our findings highlight the interplay between thermal environments and growth, underscoring the complex trade-offs amphibians face in responding to climate-related challenges.

The water-borne hormone method has been widely applied in amphibian stress studies as it does not require sacrificing individuals and provides an integrated measure of hormonal responses over a sampling period [25]. In our study, CORT release rates declined at 6 h post-treatment regardless of whether the tadpoles were initially exposed to elevated temperatures. However, this contrasts with prior findings where CORT release rates began to decline 2 h post-acute stressor and had significantly decreased by 6 h [13]. We are unsure why both our heat-treated and control tadpoles showed a decline in CORT release rates. One hypothesis is that the decrease in CORT at 6 h post-return to room temperature may, in part, be influenced by circadian rhythms. Wright et al. [37] studied late prometamorphic tadpoles and found that CORT levels exhibit diel fluctuations with CORT rhythms being responsive to the light–dark (LD) cycle. For those maintained on a 12 L:12 D cycle, as were ours, CORT decreased maximally around 1300 h and stayed low for the night. Similarly, in our study, CORT levels declined by 6 h in both treatments, which coincided with about 1600 h. This suggests that the overall suppression of CORT at this time point may reflect an underlying circadian rhythm, rather than a stress-related response in the control group. However, the significantly stronger suppression in the heat treatment compared to the control group supports our interpretation that the response in heat-exposed tadpoles represents a biologically meaningful stress effect, rather than a circadian artifact. Unlike prior studies, we did not observe elevated CORT release rates in heat-treated *I. nebulifer*. This contrasts with previous findings, where cane toads (*Rhinella marina*) exposed to daily thermal stress exhibited prolonged activation of the HPI axis with increased baseline CORT [38]. Whereas, Goff [34] reported that tadpoles housed at higher temperatures (27 °C) had lower CORT release rates and a higher body condition compared to those at cooler temperatures (19 °C), suggesting species-specific responses and context-dependent outcomes.

Our results suggest that *I. nebulifer* may exhibit physiological or behavioral adaptations that mitigate heat stress (at least for 7 days). Biogeographically, *I. nebulifer* is expanding its range and is frequently described as an urban-tolerant species [28]. This tolerance may explain why CORT release rates did not increase in response to heat exposure. Alternatively, the lack of elevated CORT could be attributed to habituation to laboratory conditions or confinement during hormone sampling [19]. However, no evidence of such confounding factors has been reported in similar studies [13]. Furthermore, our findings are consistent with prior work on amphibians, where species like the Iberian spadefoot toad (*Pelobates cultripes*) and Iberian painted frog (*Discoglossus galganoi*) did not exhibit temperature-induced changes in CORT release rates, but demonstrated faster growth under warmer conditions [12,39]. However, we found there to be long-term effects of heat, because by Day 12, significantly more tadpoles from the heat treatment were dead than the tadpoles from the lower temperature treatment group. Thus, despite minimal differences in CORT release rates, more tadpoles did not survive in the heat treatment group. Maybe the CORT release rates on Day 12 did not differ across treatments because only healthy tadpoles survived to be assayed that day. These findings suggest that while *I. nebulifer* may possess physiological or behavioral mechanisms to tolerate short-term heat exposure, prolonged thermal stress may still have significant survival costs.

By examining the physiological consequences of prolonged exposure to elevated temperatures, our study contributes to understanding how amphibians may respond to increasingly persistent thermal stress in urbanized landscapes. In regions like Texas, temperature fluctuations between day and night are often minimal due to high humidity levels, which reduce the extent of thermal cooling overnight. Moreover, Brans et al. [5] highlight that the mean temperature of urban ponds during the summer can be 3.04 °C higher than rural ponds. These urban heat island effects mean that amphibians in urban areas may have limited opportunities to escape elevated temperatures, particularly in small or isolated ponds. These findings provide insights into the potential challenges faced by amphibians in these urban habitats with limited thermal refuges.

Heat-exposed tadpoles grew significantly faster during the treatment period and continued to grow faster for five days post-treatment. This aligns with previous studies showing increased growth rates and body condition in amphibians reared at higher temperatures [40,41]. For example, *Rhinella granulosa* tadpoles exhibited greater growth at 33 °C compared to 26 °C [42]. This suggests that chronic heat exposure may enhance metabolic or physiological processes promoting growth. While stress is generally associated with reduced growth and lower body condition in amphibians [43,44], tadpoles of *I. nebulifer* from this population demonstrated the ability to sustain growth under thermal stress. This may reflect an adaptive strategy tied to its urban tolerance, where growth benefits outweigh transient endocrine adjustments. Our findings highlight the resiliency of *I. nebulifer* to chronic thermal stress [28], suggesting that amphibian species that are more tolerant of urban environments could be leveraging phenotypic plasticity to mitigate the physiological impacts of rising temperatures. This capacity likely has important implications for amphibian conservation considering continued climate change, highlighting the ability for certain species to persist in rapidly warming environments to their benefit, and possibly to the detriment of species who lack such potential. However, as noted above, there were still survival costs for a significant portion of the population. It remains unclear whether the increased growth rates in surviving tadpoles compensate for the higher mortality, potentially allowing a subset of individuals to reach metamorphosis more quickly while others succumb to thermal stress. Future research is needed to determine whether this represents a trade-off that benefits overall population persistence, or if mortality outweighs any adaptive advantages.

Despite these findings, our study has limitations. We did not measure stress responses following an additional acute stressor after the 7-day treatment exposure, which would have provided clearer insights into the recovery process. Additionally, we did not measure the snout–vent length (SVL), so we could not assess body condition. The lower CORT release rates in both treatments by 6 h may also be influenced by circadian rhythms, which we were unable to control for. Further studies are needed to evaluate whether these responses are specific to this population or *I. nebulifer* in general,, and whether they are generalizable to other species with varying thermal sensitivities, such as those from higher altitudes or latitudes. Additionally, future studies should investigate whether increased growth rates in surviving tadpoles compensate for higher mortality, potentially allowing a subset of individuals to reach metamorphosis more quickly, or whether the costs of heat-induced mortality ultimately outweigh any adaptive advantages.

## 5. Conclusions

In summary, the tadpoles of *I. nebulifer* exposed to a week of elevated temperatures showed no significant increase in CORT release rates but exhibited a stronger suppression at 6 h post-return to room temperature compared to the controls, suggesting a distinct physiological response to heat. However, water-borne CORT measures alone could not confirm whether tadpoles experienced chronic stress from extended heat exposure. Notably, heat-exposed tadpoles grew faster, suggesting a potential trade-off between stress recovery and growth. However, this accelerated growth came at a cost, as significantly more tadpoles from the heat treatment died by Day 12 compared to the controls. This suggests that while tadpoles of *I. nebulifer* may employ physiological or behavioral mechanisms to tolerate heat exposure, prolonged thermal stress still imposes significant survival costs. It is also possible that CORT release rates did not differ between the treatments on Day 12 because only the healthiest tadpoles survived to be assayed. Unfortunately, we were unable to determine the lag time between HPI axis activation and the hormonal response to chronic stress using non-invasive sampling methods. The absence of clear HPI axis activation in heat-exposed tadpoles may be due to habituation or the limitations of this method in detecting hormonal responses to prolonged stress. Future studies could adopt a comparative approach, incorporating species with lower thermal tolerance or testing responses across varying habitat conditions, to better understand how amphibians physiologically and behaviorally navigate environmental temperature fluctuations in a rapidly changing world.

## Figures and Tables

**Figure 1 biology-14-00255-f001:**
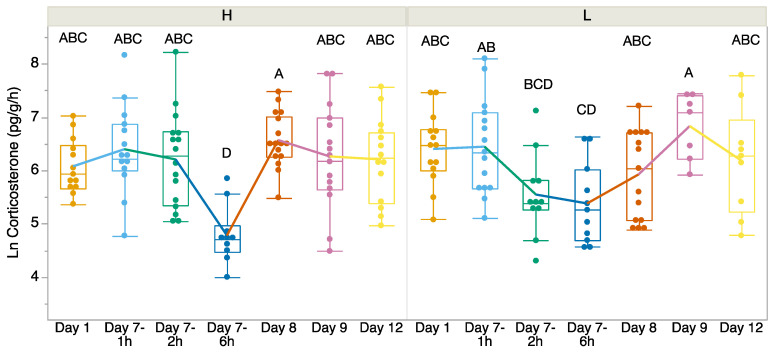
Corticosterone release rates (pg/g/h) over time from Day 1 (baseline 23 °C) before exposure, at which point *Incilius nebulifer* tadpoles were exposed to higher (H) water temperature (32 °C) or lower (L) water temperature (23 °C) for 7 days. On Day 8, the tadpoles were returned to the same water temperature (23 °C). The boxplot at each time point explains the distribution of data, while the points represent individual observations. Different letters indicate significant differences.

**Figure 2 biology-14-00255-f002:**
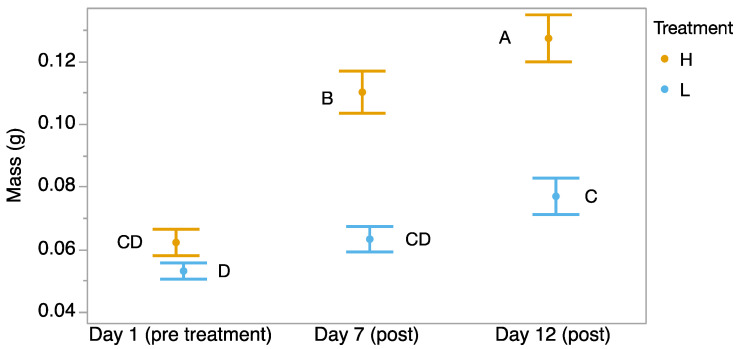
Mass (g) of *Incilius nebulifer* tadpoles over time starting from before exposure, to 32 °C water, to the end of 7 days of exposure, and after 5 more days when both treatments were maintained at 23 °C. Different letters indicate significant differences.

## Data Availability

Data will be made available upon request.

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
