# Peer review of "Investigating the Temporal Effects of Thermal Stress on Corticosterone Release and Growth in Toad Tadpoles"

_biology, 2025, doi:10.3390/biology14030255_

Round 1
Reviewer 1 Report
Comments and Suggestions for Authors
Review of: Exploring Stress Recovery and Growth in Gulf Coast Toads Using 2 Water-Borne Hormone Methods
Authors: Saeid Panahi Hassan Barough, Dillon J. Monroe, Thomas C. Clark & Caitlin R. Gabor
Summary: The authors investigated the effects of temperature on growth and corticosterone (CORT) release rates of Gulf Coast Toad tadpoles and found that temperature affected growth and survival, but not necessarily CORT release rates, only their time to rebound to baseline. The paper is generally well written but I have questions and potential concerns about the experimental design and inferences from the paper. For this reason, I suggest major revisions until more clarity is added.
Major comments:
11 My first and major caveat is that the experiment and collection protocol seem flawed, but I could just be confused. The authors seek to understand CORT responses of this species to elevations in temperature. However, and importantly, they exposed animals to high temperatures but then collected CORT at room temperatures and thus the CORT response they are seeing is not to high temperatures per say, but to going from high to low temperatures. Even the low temperature animals had a dip in CORT during collection which also suggests that animals were responding to the collection procedures or changes in housing or changes in temperature DURING collection (e.g., H to room temp for collection or L to room temp for collection). For the L group, were they held at room temperature (room temperature was not stated)? If so then …
22 The stressors that H v. L groups are exposed to are NOT equivalent. H group goes from 32 to room temperature, L group goes from 23 to room temperature. Those are not equivalent so we would expect differences based on the magnitude of stressors alone compared to being exposed to a higher temperature for a few days. Nevertheless, they both respond (a dip in CORT), but have different rebounds (back to baseline). This suggests its not their CORT responses to temperature, but a response to being put in a collection chamber at the same or different temperatures so the effect of temperature is confounded with the effect of collection. This is a major caveat if true. It would have probably been better to hold temperature constant (collect CORT in H temp for H and L temp for L) and use a stressor that was equivalent (e.g., shaking or handling). This should be discussed in the caveat section of the manuscript in more detail (see minor comments, below).
33 I also don’t see the ecological significance of this experiment when animals can move from higher to lower temperatures regularly, hourly, on a day to day basis when temperatures warm and cool and when animals move to shallower areas etc. This needs to be discussed. Do the authors think this is ecologically relevant?
44 Statistical inference. The authors do not specify what they collected. One egg mass? Some larvae? From one Pond? From several ponds? Yet they go on to expand inferences to the entire species. This is dangerous. The authors need to cut their inference to whatever unit they sampled (one egg mass or pond) because they don’t know how results would differ if they sampled across the region, population, or species. Be sure to make these changes in the discussion.
55 A consensus chronic stress profile does not exist so how do we know what it looks like? Given that amphibians occur in heterogeneous environments they could experience 32 C water commonly. The authors need to be careful about ‘chronic stress’ and temper those discussions. A discussion of heterogenous environments and microhabitats might be useful in the discussion to temper this.
Dickens, Molly J., and L. Michael Romero. "A consensus endocrine profile for chronically stressed wild animals does not exist." General and comparative endocrinology 191 (2013): 177-189.
66 Analytical and biological validation should be done for each species. Has it been done for this species? If so, cite those papers. If not, it would be important to see demonstrated parallelism, cold spikes, etc.
Minor comments:
Abstract
L12 CORT metabolites or CORT? Will need to clarify based on methods. Could end up being CORT AND CORT metabolites depending on the assays/EIAs. Most of these EIAs do pick up some metabolites so specify that near the EIA section if that’s what you mean.
L30 stress-recovery despite no changes in CORT? Unclear.
L32 In other studies?
L33 Is it resilience if most amphibians grow faster in warmer environments?
Introduction
L44-46 citations are doubled. Remove doubles.
L48 urban heat island might need more explanation for lay readers
L62 while that’s a general citation about the HPA/I axis, its not a citation about effects of temperature on CORT. Maybe provide another or another sentence showing evidence that temperature influences CORT responses.
L72 varied baseline AND STRESS-INDUCED CORT levels …
L76-79 ‘better’ is very objective without context, please provide context. Other measures can be ‘better’ because they are NOT integrated/across time or space depending on the circumstance/context.
L89 Why the ‘Z’ before this author?
L117 more context required. What did the authors posit as to why this occurs? How would that relate to survival?
L121 I don’t see how this last sentence relates to the one before or how it transitions to the next paragraph. Could remove.
L128 what’s the evidence that they are adapted to variable thermal environments?
L131 given your sampling protocol (2h, 6h, 24h …) wouldn’t you want immediate hormonal responses? Although waterborne is integrated, its typically only ‘integrated’ over the sampling time (how long they ‘soaked’ for hormone collection). If you include this here it should be better explained.
L134 and before, these are more predictions than hypotheses.
Materials and Methods
L144 more context needed. All from one clutch? All from one pond? How did this pond differ from other habitats the species occurs in? This has important ramifications for the inferences you can draw (to the population, pond, or species).
L162 on or one? Also, when/how were water changes done? Important detail.
L167 What was room temperature? 23? 21? Important detail given you collect at this temperature.
L168 Why allow to return to room temperature? Then you’re getting hormone responses to acclimation to room temperature after a high temperature spike? That alone could be a shock/stressor per say. But this would be different magnitude stressor for H or L groups so again you are not measuring temperature tolerance but rather their response to a return to room temperatures (or not).
L186 has this protocol been analytically or biologically validated for this species? It would be important to show information on parallelism of samples to standard curve, recovery, cold spikes, etc. A handling challenge would also be important to show you can even detect changes after an acute stressor.
Results
L198 I am still struggling with the ecological significance of this experiment. You expose them to higher temperatures and then expose them to room temperatures again … both had reductions in CORT release rates, just at different times. This is assuming that animals occur in homogenous environments when they do not. Environments are heterogeneous and amphibian larvae often seek out warmer, shallower areas and retreat to cooler deeper areas when threatened. This deserves more discussion later.
Figure 1 – it seems to me that if room temperature is 23 and you saw changes in the L (23C) group then you might just be seeing a response to the hormone collection v. a response to changes in temperature (e.g., moving from H to room temperature/L). Important for discussion.
Discussion
L225 Aren’t most animals adapted to variable thermal environments?
L229 “allowing us to assess the persistence of elevated CORT and recovery time as individuals re-established homeostasis.” I still don’t really agree with this. You had them in warmer or cooler environments then moved them to room temperature. If you wanted to see the persistence of elevated CORT and recovery time then you should have collected hormones at high temperature to avoid them responding to a CHANGE in temperature. I feel like this is a flawed experiment, which is fine but needs to be discussed that they are responding to changes in temperature (32 to 23) during collection. A CORT response can be initiated in minutes (3-5 usually) and this can be detected that quickly too.
L231 see above did you find that chronic exposure delayed this? I guess maybe so. They were exposed to H or L, then exposed to room temp during collection, yet the low returned to baseline faster? Yes maybe so, but the stressor was not equivalent between H and L groups. The H group went from a high to low water temperature whereas they L group was unchanged so we would expect they would have a lesser response. A better thing here may have been a handling challenge during collection in their respective water temperatures.
L234 this is not particularly interesting given that we have known for decades that growth rates of animals, particularly ectotherms, are dependent on temperature.
L261 to 263 I feel like this is confounding a generalist species (urban tolerant) with thermal tolerance? That’s not always the case, this needs to be clarified.
L295 yes, this. This is an important caveat.
L295-296 talk more about this. How could it have done this? Also talk about how the H v. L were not exposed to an equivalent stressor during CORT collection (see above).
L301 the other important caveat is statistical inference. It was not clear in your methods about whether the amphibians were collected from one clutch, one pond, one general location? If its one clutch then your inferences are not even generalizable to the species, but only to that clutch or pond. That needs to be made clear here as a caveat.
L304-305 Did they? Or did it just appear that way compared to the L group because they were exposed to stressors of different magnitude?
L306 Clarify to say “this population of I. nebulifer” but really any two animals exposed to different stressors would be expected to respond differently so is that actually phenotypic plasticity?
Author Response
Please find our responses bulleted below and track changes in the document
Major comments:
11 My first and major caveat is that the experiment and collection protocol seem flawed, but I could just be confused. The authors seek to understand CORT responses of this species to elevations in temperature. However, and importantly, they exposed animals to high temperatures but then collected CORT at room temperatures and thus the CORT response they are seeing is not to high temperatures per say, but to going from high to low temperatures. Even the low temperature animals had a dip in CORT during collection which also suggests that animals were responding to the collection procedures or changes in housing or changes in temperature DURING collection (e.g., H to room temp for collection or L to room temp for collection). For the L group, were they held at room temperature (room temperature was not stated)? If so then …
- We understand your concern. Our primary goal was to understand how the glucocorticoid profile changed (or didn’t) within hours of the temperature being decreased after the initial exposure to heat (H) as compared to the control (L). Previously, on line 148, we noted that room temperature was 23C but obviously we need to make it more clear. We have now reiterated that 23C is room temperature when we first explained the design.
As for the dip - The “dip” occurred 6 hours after they had already reached room temperature and were placed in the collection cups so this does not indicate that the collection per se caused the dip. We might expect that in the first hour because we know from prior studies that water-borne CORT can respond to stressors with an hour.
22 The stressors that H v. L groups are exposed to are NOT equivalent. H group goes from 32 to room temperature, L group goes from 23 to room temperature. Those are not equivalent so we would expect differences based on the magnitude of stressors alone compared to being exposed to a higher temperature for a few days. Nevertheless, they both respond (a dip in CORT), but have different rebounds (back to baseline). This suggests its not their CORT responses to temperature, but a response to being put in a collection chamber at the same or different temperatures so the effect of temperature is confounded with the effect of collection. This is a major caveat if true. It would have probably been better to hold temperature constant (collect CORT in H temp for H and L temp for L) and use a stressor that was equivalent (e.g., shaking or handling). This should be discussed in the caveat section of the manuscript in more detail (see minor comments, below).
- In this study only the H was brought to room temperature because L was at room temperature already. The dip in cort was sooner in the Low temperature treatment than the high temp treatment so this does not in and of itself indicate that it is due to the stress of collection. Further, in Forsburg et al. (2019) using tadpoles we found that cort values are repeatable over repeated measures for up to 6h hours indicating that the collection method minimally stressed the tadpoles. We agree that in retrospect, it would have been more useful to additionally stress (shaking) both the L and H treatment tadpoles before measuring their recovery-, but we were thinking of the H treatment as a stressor so didn’t including shaking. We had noted this in the discussion. But, we still think that first getting tadpoles from the Heat treatment to room temperature provides a better comparison because in some species of amphibians CORT is higher at higher temperatures so we wanted to compare their response at the same temperature and then any difference would have been due to the 7 days of exposure to heat.
33 I also don’t see the ecological significance of this experiment when animals can move from higher to lower temperatures regularly, hourly, on a day to day basis when temperatures warm and cool and when animals move to shallower areas etc. This needs to be discussed. Do the authors think this is ecologically relevant?
- We appreciate your concern about the ecological relevance of our experiment. While our study does not directly mimic the day-to-day or hourly temperature fluctuations that amphibians might experience in their natural habitats, it serves as a starting point for understanding physiological responses under controlled conditions. This controlled approach allows us to isolate the effects of sustained elevated temperatures, which are increasingly relevant in the context of climate change and urbanization.
- We have now added the following paragraph to the discussion to further address this comment “By examining the physiological consequences of prolonged exposure to elevated temperatures, our study contributes to understanding how amphibians may respond to increasingly persistent thermal stress in urbanized landscapes. In regions like Texas, temperature fluctuations between day and night are often minimal due to high humidity levels, which reduce the extent of thermal cooling overnight. Moreover, Brans et al. (2018) highlight that the mean temperature of urban ponds during summer can be 3.04°C higher than rural ponds. These urban heat island effects mean that amphibians in urban areas may have limited opportunities to escape elevated temperatures, particularly in small or isolated ponds. These findings provide insights into the potential challenges faced by amphibians in these urban habitats with limited thermal refuges.”
44 Statistical inference. The authors do not specify what they collected. One egg mass? Some larvae? From one Pond? From several ponds? Yet they go on to expand inferences to the entire species. This is dangerous. The authors need to cut their inference to whatever unit they sampled (one egg mass or pond) because they don’t know how results would differ if they sampled across the region, population, or species. Be sure to make these changes in the discussion.
- These were collected as tadpoles, not eggs and from one pond (which we noted by including the gps location). We assume that these tadpoles came from many different individuals given that the pond was not small and the tadpoles were not all the same size. But we agree that we need to be more careful to only infer what we know for this population and have tried to make those edits in the discussion.
- On line 261 we added the words in italics to this sentence “Our results suggest that I nebulifer, from this population, may exhibit”. We made a similar change on line 298 and line 322. Finally ,we added the words in italics on line 315 Further studies are needed to evaluate whether these responses are specific to this population or nebulifer in general and or are generalizable to other species with varying thermal sensitivities
55 A consensus chronic stress profile does not exist so how do we know what it looks like? Given that amphibians occur in heterogeneous environments they could experience 32 C water commonly. The authors need to be careful about ‘chronic stress’ and temper those discussions. A discussion of heterogenous environments and microhabitats might be useful in the discussion to temper this.\
- We appreciate this important point. Our revised discussion now tempers the use of "chronic stress" by emphasizing the need for caution in interpreting prolonged temperature exposure as a definitive chronic stressor. We have now edited our last two sentences to say
“While this non-invasive approach provides a valuable tool for assessing integrated stress responses to prolonged temperature exposure, we could not determine whether it takes longer to observe the effects of chronic stress or how long recovery truly requires following extended heat exposure. Future studies could adopt a comparative approach , incorporating species with lower thermal tolerance or testing responses across varying habitat conditions, to better understand how amphibians physiologically and behaviorally navigate environmental temperature fluctuations in a rapidly changing world.”
66 Analytical and biological validation should be done for each species. Has it been done for this species? If so, cite those papers. If not, it would be important to see demonstrated parallelism, cold spikes, etc.
- We have validated this method in many species including this one. See review by Narayan et al. (2019) where we summarize the many papers validating this method in many species.
-
Minor comments:
Abstract
L12 CORT metabolites or CORT? Will need to clarify based on methods. Could end up being CORT AND CORT metabolites depending on the assays/EIAs. Most of these EIAs do pick up some metabolites so specify that near the EIA section if that’s what you mean.
- We added that CORT (and metabolites) were measured with Cayman Kits. But not in the Abstract since it should be short.
L30 stress-recovery despite no changes in CORT? Unclear.
- Now that we added the chi sq test for survival and found lower survival for heat treated tadpoles, we have edited this sentence to say “Heat-exposed tadpoles also showed significantly faster growth during and after treatment but lower survival to 12days, indicating a potential trade-off between survival and accelerated growth”
L32 In other studies?
- We are not sure what your question is here.
L33 Is it resilience if most amphibians grow faster in warmer environments?
- We changed it to “Overall, our study highlights a trade-off for populations of nebulifer when exposed to thermal stress”
Introduction
L44-46 citations are doubled. Remove doubles.
- fixed
L48 urban heat island might need more explanation for lay readers
- We reworded the first time it comes up and in the last paragraph.
L62 while that’s a general citation about the HPA/I axis, its not a citation about effects of temperature on CORT. Maybe provide another or another sentence showing evidence that temperature influences CORT responses.
- This was intended to be a broad sentence because later in the introduction we specifically address this topic with references
L72 varied baseline AND STRESS-INDUCED CORT levels …
- Added
L76-79 ‘better’ is very objective without context, please provide context. Other measures can be ‘better’ because they are NOT integrated/across time or space depending on the circumstance/context.
- Good point. We removed the word better
L89 Why the ‘Z’ before this author?
- Endnote was being weird. We fixed it.
L117 more context required. What did the authors posit as to why this occurs? How would that relate to survival?
- We added “The authors hypothesized that lower CORT levels prolonged the time to metamorphosis, allowing tadpoles to grow larger before”
L121 I don’t see how this last sentence relates to the one before or how it transitions to the next paragraph. Could remove.
- Good point -removed
L128 what’s the evidence that they are adapted to variable thermal environments?
- We added more details to support this “Incilius nebulifer is native to central and eastern Texas and northeast Mexico but has been expanding its range northward and eastward in the United States ((Mendelson Iii et al., 2015)). This range expansion has been strongly associated with urbanized and disturbed habitats, suggesting that nebulifer may be particularly urban-tolerant (Monroe et al., 2024). Given that urban areas experience elevated temperatures due to the heat island effect, this species provides a valuable model for understanding how amphibians physiologically respond to chronic heat exposure in increasingly warming and urbanized environments.”
L131 given your sampling protocol (2h, 6h, 24h …) wouldn’t you want immediate hormonal responses? Although waterborne is integrated, its typically only ‘integrated’ over the sampling time (how long they ‘soaked’ for hormone collection). If you include this here it should be better explained.
- Hmm, I think integrated also mean integrated over the time it takes to be metabolize and therefore is not just limited to one hour, but potentially up to a few days, which is why we were asking this question.
L134 and before, these are more predictions than hypotheses.
Materials and Methods
L144 more context needed. All from one clutch? All from one pond? How did this pond differ from other habitats the species occurs in? This has important ramifications for the inferences you can draw (to the population, pond, or species).
L162 on or one? Also, when/how were water changes done? Important detail.
- That was at line 150 – every 5 days before we started but once they were in the deli cup not until we took the first CORT measure. We added that in.
L167 What was room temperature? 23? 21? Important detail given you collect at this temperature.
- We included this information at the start of the methods on line 148 – 23C in the first draft and now we have added it in later with the experimental methods
L168 Why allow to return to room temperature? Then you’re getting hormone responses to acclimation to room temperature after a high temperature spike? That alone could be a shock/stressor per say. But this would be different magnitude stressor for H or L groups so again you are not measuring temperature tolerance but rather their response to a return to room temperatures (or not).
- In this study only the Heat treatment was brought to room temperature because the L was at room temperature already. We didn’t want to measure how heat at that moment affected CORT but the integrated response from one week of exposure to high temperatures.
L186 has this protocol been analytically or biologically validated for this species? It would be important to show information on parallelism of samples to standard curve, recovery, cold spikes, etc. A handling challenge would also be important to show you can even detect changes after an acute stressor.
- See above. We have many papers published using these methods and those are found in the Narayan et al. (2019) review. Many more papers have been published since then.
Results
L198 I am still struggling with the ecological significance of this experiment. You expose them to higher temperatures and then expose them to room temperatures again … both had reductions in CORT release rates, just at different times. This is assuming that animals occur in homogenous environments when they do not. Environments are heterogeneous and amphibian larvae often seek out warmer, shallower areas and retreat to cooler deeper areas when threatened. This deserves more discussion later.
- We appreciate the concern about the ecological relevance of our experiment. While our study does not directly mimic the day-to-day or hourly temperature fluctuations that amphibians might experience in their natural habitats, it serves as a starting point for understanding physiological responses under controlled conditions. This controlled approach allows us to isolate the effects of sustained elevated temperatures, which are increasingly relevant in the context of climate change and urbanization. Reviewer one also asked about this and we made a few changes to further address this.
Figure 1 – it seems to me that if room temperature is 23 and you saw changes in the L (23C) group then you might just be seeing a response to the hormone collection v. a response to changes in temperature (e.g., moving from H to room temperature/L). Important for discussion.
- The change in the low treatment occurred at 2hrs, where as it occurred 6 hours after the high treatment had already reached room temperature so to me this does not indicate that the collection method alone caused the dip. Further, in Forsburg et al. (2019) using tadpoles we found that cort values are repeatable over repeated measures for up to 6h hours indicating that the collection method minimally stressed the tadpoles. In sum, we do not agree that these results were due to the stress of collection.
Discussion
L225 Aren’t most animals adapted to variable thermal environments?
- Not that we know of or climate change wouldn’t be as big of a hit.
L229 “allowing us to assess the persistence of elevated CORT and recovery time as individuals re-established homeostasis.” I still don’t really agree with this. You had them in warmer or cooler environments then moved them to room temperature. If you wanted to see the persistence of elevated CORT and recovery time then you should have collected hormones at high temperature to avoid them responding to a CHANGE in temperature. I feel like this is a flawed experiment, which is fine but needs to be discussed that they are responding to changes in temperature (32 to 23) during collection. A CORT response can be initiated in minutes (3-5 usually) and this can be detected that quickly too.
- Yes, but if we were truly able to measure a change in CORT due to the chronic exposure to Heat we felt that we had to keep them at high temperatures but then not measure cort at the high temp to make it comparable to low temp since higher temps in and of itself may raise cort. We brought the temp down slowly to avoid a shock.
L231 see above did you find that chronic exposure delayed this? I guess maybe so. They were exposed to H or L, then exposed to room temp during collection, yet the low returned to baseline faster? Yes maybe so, but the stressor was not equivalent between H and L groups. The H group went from a high to low water temperature whereas they L group was unchanged so we would expect they would have a lesser response. A better thing here may have been a handling challenge during collection in their respective water temperatures.
- We agree that also shaking them would have been better just before our first day 7 measure but sadly we didn’t think of it. We already noted this in the discussion.
L234 this is not particularly interesting given that we have known for decades that growth rates of animals, particularly ectotherms, are dependent on temperature.
- yes
L261 to 263 I feel like this is confounding a generalist species (urban tolerant) with thermal tolerance? That’s not always the case, this needs to be clarified.
- Urban tolerance requires tolerance to many environmental changes. In the intro to explain this we added “Given that urban areas experience elevated temperatures due to the heat island effect, this species provides a valuable model for understanding how amphibians physiologically respond to chronic heat exposure in increasingly warming and urbanized environments.” L155-159 to specify we are looking at the temperature effect of urbanization.
L295 yes, this. This is an important caveat.
- thanks
L295-296 talk more about this. How could it have done this? Also talk about how the H v. L were not exposed to an equivalent stressor during CORT collection (see above).
- We don’t agree, because the point was to bring them down slowly to room temperature, such that the only stressor was the 7 days of heat.
L301 the other important caveat is statistical inference. It was not clear in your methods about whether the amphibians were collected from one clutch, one pond, one general location? If its one clutch then your inferences are not even generalizable to the species, but only to that clutch or pond. That needs to be made clear here as a caveat.
- We added for this population in the references to avoid the inference being applied to the whole species. we collected tadpoles from a pond so they were likely from more than one clutch.
L304-305 Did they? Or did it just appear that way compared to the L group because they were exposed to stressors of different magnitude?
- This is our interpretation.
L306 Clarify to say “this population of I. nebulifer” but really any two animals exposed to different stressors would be expected to respond differently so is that actually phenotypic plasticity?
- Yes, we added that in too.
Reviewer 2 Report
Comments and Suggestions for Authors
While this study "Exploring Stress Recovery and Growth in Gulf Coast Toads Using Water-Borne Hormone Methods" presents a novel approach, several critical aspects of the experimental design require further consideration.
Major concerns include:
1. The experimental tadpole density (3 individuals/L) raises questions about potential density-dependent effects: Has the author evaluated whether this density introduces confounding density-related stress? How does this density compare to natural population densities in the field?
2. The selected experimental temperatures (23°C and 32°C) require additional justification: What is the ecological relevance of these specific temperature points? How do these values relate to the species' natural thermal environment?
3. Most crucially, the fundamental methodological concern centers on the water-borne hormone analysis: Has the validity of this hormone measurement technique been cross-validated using established methodologies? What evidence supports the reliability and accuracy of the water-borne hormone measurement approach?
Specific comments:
In Abstract:
Please provide a clear rationale for using CORT as a stress indicator in amphibians, including its biological significance and advantages as a metric.
In Introduction:
1. The first paragraph discussing global warming impacts is somewhat scattered. Consider reorganizing to more specifically highlight temperature effects on amphibians.
2. References in Lines 44-45 are duplicated and should be consolidated.
3. Lines 126-141: The rationale for selecting I. nebulifer as the study species needs to be better justified. What makes this species particularly suitable for addressing the research questions?
In Materials and Methods:
1. Basic information about the developmental stages and body mass of collected tadpoles should be included.
2. Lines 144-164: Please provide density information for both acclimation and experimental periods. High density can induce stress responses, making it difficult to attribute CORT changes specifically to temperature effects.
3. Lines 166-172: The sampling time points need better justification. What was the rationale behind selecting these specific intervals?
In Results:
1. The variation in sample sizes across different sampling times needs explanation in the Methods section.
2. Figure 1's placement should be reconsidered for better flow of the manuscript.
In Discussion:
The limitations section should be expanded to include more detailed discussion of both experimental design constraints and technical limitations.
Author Response
Please find our responses to each comment bulleted and the changes were tracked in our MS.
Major concerns include:
- The experimental tadpole density (3 individuals/L) raises questions about potential density-dependent effects: Has the author evaluated whether this density introduces confounding density-related stress? How does this density compare to natural population densities in the field?
- I am not sure where the reviewer saw this. We started out with tadpoles in two 6L containers and then individually maintained them in 470 ml deli cups (see line 152). This density is similar to that used in many other experiments and the field.
- The selected experimental temperatures (23°C and 32°C) require additional justification: What is the ecological relevance of these specific temperature points? How do these values relate to the species' natural thermal environment?
- In Monroe et al. (2024), using Worldclim data, they found that the annual temperature for some of the southern populations of I nebulifer is 23.2C. The high temperature was as high as we could get inexpensive tank heaters to go and within the range that this species may experience. We have measured water temperatures as high as 34C in central Texas.
- Most crucially, the fundamental methodological concern centers on the water-borne hormone analysis: Has the validity of this hormone measurement technique been cross-validated using established methodologies? What evidence supports the reliability and accuracy of the water-borne hormone measurement approach?
- Yes, the water-borne hormone method has been validated many times. We published the following paper citing all the papers that have validated the methods and reference them in this review - Narayan, E. J., Forsburg, Z. R., Davis, D. R. & Gabor, C. R. Non-invasive Methods for Measuring and Monitoring Stress Physiology in Imperiled Amphibians. Frontiers in Ecology and Evolution 7, doi:10.3389/fevo.2019.00431 (2019).
- We also added the following sentence on line 169 “The water-borne CORT collection method has been validated for this species and many others amphibian species (Narayan et al., 2019)”.
Specific comments:
In Abstract:
Please provide a clear rationale for using CORT as a stress indicator in amphibians, including its biological significance and advantages as a metric.
- We added the following sentence “Corticosterone (CORT) is a key glucocorticoid hormone that regulates energy balance and physiological responses to environmental stressors, making it a valuable biomarker for assessing how organisms cope with changing conditions.”
In Introduction:
- The first paragraph discussing global warming impacts is somewhat scattered. Consider reorganizing to more specifically highlight temperature effects on amphibians.
- We have edited two sentences, mostly, to read “Among these threats, rising temperatures due to climate change and urbanization pose a significant challenge to aquatic ecosystems. Land use changes and the urban heat island effect have increased water temperatures by up to 5°C in some areas of the northeastern United States, with documented negative effects on fish and amphibians (Brans et al., 2018; Chapman et al., 2017; Gartland, 2012). Warmer water accelerates metabolic rates in aquatic organisms, leading to cascading effects on immune function, locomotor performance, and behavior (Brans et al., 2018; Rollins-Smith & Le Sage, 2023; Taylor et al., 2021; Winterová, 2021)”
- References in Lines 44-45 are duplicated and should be consolidated.
- Fixed
- Lines 126-141: The rationale for selecting I. nebulifer as the study species needs to be better justified. What makes this species particularly suitable for addressing the research questions?
- Here we added the following sentences “ nebulifer is native to central and eastern Texas and northeast Mexico but has been expanding its range northward and eastward in the United States (Mendelson et al., 2015). This range expansion has been strongly associated with urbanized and disturbed habitats, suggesting that I. nebulifer may be particularly urban-tolerant (Monroe et al., 2024). Given that urban areas experience elevated temperatures due to the heat island effect, this species provides a valuable model for understanding how amphibians physiologically respond to chronic heat exposure in increasingly warming and urbanized environments.”
In Materials and Methods:
- Basic information about the developmental stages and body mass of collected tadpoles should be included.
- We added “Tadpoles weighed between 34-93 mg at this stage and were all in Gosner stages 30-35 (Gosner, 1960)”
- Lines 144-164: Please provide density information for both acclimation and experimental periods. High density can induce stress responses, making it difficult to attribute CORT changes specifically to temperature effects.
- Tadpoles were individually maintained in 470ml deli cups which was on line 152.
- Lines 166-172: The sampling time points need better justification. What was the rationale behind selecting these specific intervals?
- We added this “Our short-term sampling time points were based on methods from a similar study examining acute stress responses (Forsburg et al., 2019), and we extended the sampling period to assess potential longer-term changes in corticosterone release rates”
In Results:
- The variation in sample sizes across different sampling times needs explanation in the Methods section.
- We added “Owing to removing some samples with high CV and a few lost to spills, our sample sizes vary across sampling time points.”
- Figure 1's placement should be reconsidered for better flow of the manuscript.
- We moved figure one down.
In Discussion:
The limitations section should be expanded to include more detailed discussion of both experimental design constraints and technical limitations.
- We have expanded our discussion to consider constraints and limitations
Reviewer 3 Report
Comments and Suggestions for Authors
The manuscript "Exploring Stress Recovery and Growth in Gulf Coast Toads Using Water-Borne Hormone Methods" presents a well-designed and relevant study focusing on the impact of chronic thermal stress on Gulf Coast toad tadpoles (Incilius nebulifer). Using non-invasive water-borne hormone methods, the authors examined corticosterone (CORT) release dynamics and growth responses, providing insights into the physiological and ecological adaptations of amphibians to thermal stress. The results suggest that while chronic heat exposure slightly delayed CORT recovery, it promoted faster growth, highlighting potential trade-offs in stress recovery and growth. This study holds significant potential for understanding how amphibians, particularly thermally tolerant species, adapt to environmental challenges associated with climate change and urbanization. The research is very meaningful, but I have a few questions and suggestions:
Major Comments:
1. Why was 32°C selected for the high-temperature group? Was a semi-lethal temperature test conducted to determine its relevance?
2. The experiment spanned seven days, but data only include Day 7 and beyond. Daily monitoring would better show trends during the week.
3.The small sample size reduces representativeness
4. Large CORT fluctuations at the 6-hour mark on Day 7 might be linked to circadian rhythms. Does this suggest significant 24-hour variation?
Minor Comments:
1.Improve figure formatting and clarify legends, such as what "different letters" represent statistically.
2.Provide mortality data to offer a complete picture of tadpole tolerance.
3.Include details on how body weight was measured and its frequency for clarity.
Author Response
Please find our responses to each point bulleted and the changes tracked in our MS
- Why was 32°C selected for the high-temperature group? Was a semi-lethal temperature test conducted to determine its relevance?
- We have now added the following to the methods “Monroe et al. [28], using Worldclim data, found that the annual temperature for some of the southern populations of nebulifer is 23.2C so we used that as our lower value and we have measured temperatures as high as 34C in Central Texas in ponds with tadpoles so we set the upper limit below this”
- The experiment spanned seven days, but data only include Day 7 and beyond. Daily monitoring would better show trends during the week.
- Our primary research question examined how 'longer term' stressor exposure affects water-borne corticosterone measures. We specifically avoided interrupting the stressor during the 7-day period, with feeding being the only interaction with the tadpoles until we measured the water-borne hormones on the 7th
3.The small sample size reduces representativeness
- We have revised the paper to now state: 'Sample sizes ranged from 7-16 individuals per time point and treatment owing to removing some samples with high CV and mortality that occurred after day 7 (11 died by day 12).' (lines 198-201).
While our day 9 sample was smaller than others, our results did not hinge on that particular time point. Overall, we argue that our sample sizes were sufficient to maintain representativeness and do not limit the validity of our conclusions.
- Large CORT fluctuations at the 6-hour mark on Day 7 might be linked to circadian rhythms. Does this suggest significant 24-hour variation?
- We had mentioned this on line 344-345 but added “by 6 hour “instead of later in the day to make it more clear.
Minor Comments:
1.Improve figure formatting and clarify legends, such as what "different letters" represent statistically.
- We are not sure what is wrong with our figure formatting. As for the use of “different letters” -we are following traditional wording. This notation means that data points or groups labeled with different letters (like a, b, c or A, B, C) are statistically significantly different from each other, while those sharing the same letter are not significantly different. Here different letters indicate significant differences based on alpha <0.05.
2.Provide mortality data to offer a complete picture of tadpole tolerance.
- We only obtained mortality up through day 12. We had not planned on recording mortality after day 12 and we did not take the necessary data to perform a survival analysis. From the data we did take we determined that “significantly more tadpoles in the heat treatment were dead by day 12 (Pearson test, χ² =6.42, p=0.013)” on line 236.”
3.Include details on how body weight was measured and its frequency for clarity.
- On line 183-184 we already note that “After each hormone sampling day (3 days), we measured their body mass.” These are the only times we measured mass to decrease handling
3.Include details on how body weight was measured and its frequency for clarity.
- On line 207 we note that we measured body mass on each hormone sampling day.
Round 2
Reviewer 2 Report
Comments and Suggestions for Authors
Please revise the title to replace 'Gulf Coast Toads' with 'toad tadpoles' to better reflect the developmental stage studied, and add growth-related terms to the keywords to fully represent your research scope on stress recovery and growth using Water-Borne Hormone Methods.
Please note that Biology-Basel follows the numerical citation system. All in-text citations should use square brackets with numbers (e.g., [1] for single references, [2,3] for multiple references, or [1-4] for consecutive reference ranges). The manuscript should be revised accordingly to ensure all citations conform to this format.
Author Response
Please revise the title to replace 'Gulf Coast Toads' with 'toad tadpoles' to better reflect the developmental stage studied, and add growth-related terms to the keywords to fully represent your research scope on stress recovery and growth using Water-Borne Hormone Methods.
- We changed the title to “toad tadpoles” as suggested. Because growth is in the title we didn’t want to repeat the same word so we added development to the keywords and thermal stress
Please note that Biology-Basel follows the numerical citation system. All in-text citations should use square brackets with numbers (e.g., [1] for single references, [2,3] for multiple references, or [1-4] for consecutive reference ranges). The manuscript should be revised accordingly to ensure all citations conform to this format.
- We were waiting for closer to a final version to change over to numbering our references.